# Targeting Engineered Nanoparticles for Breast Cancer Therapy

**DOI:** 10.3390/pharmaceutics13111829

**Published:** 2021-11-01

**Authors:** Kumar Ganesan, Yan Wang, Fei Gao, Qingqing Liu, Chen Zhang, Peng Li, Jinming Zhang, Jianping Chen

**Affiliations:** 1Li Ka Shing Faculty of Medicine, School of Chinese Medicine, The University of Hong Kong, Hong Kong, China; kumarg@hku.hk (K.G.); wangyans@hku.hk (Y.W.); u3006022@connect.hku.hk (Q.L.); 2School of Pharmacy, Chengdu University of Traditional Chinese Medicine, Chengdu 611137, China; feigao207@yeah.net (F.G.); 1990@126.com (C.Z.); 3Shenzhen Institute of Research and Innovation, The University of Hong Kong, Shenzhen 518063, China; 4State Key Laboratory of Quality Research in Chinese Medicine, Institute of Chinese Medical Sciences, University of Macau, Macao 999078, China; pengli@um.edu.mo

**Keywords:** nanoparticles, ligands, engineering, therapeutic effects, breast cancer

## Abstract

Breast cancer (BC) is the second most common cancer in women globally after lung cancer. Presently, the most important approach for BC treatment consists of surgery, followed by radiotherapy and chemotherapy. The latter therapeutic methods are often unsuccessful in the treatment of BC because of their various side effects and the damage incurred to healthy tissues and organs. Currently, numerous nanoparticles (NPs) have been identified and synthesized to selectively target BC cells without causing any impairments to the adjacent normal tissues or organs. Based on an exploratory study, this comprehensive review aims to provide information on engineered NPs and their payloads as promising tools in the treatment of BC. Therapeutic drugs or natural bioactive compounds generally incorporate engineered NPs of ideal sizes and shapes to enhance their solubility, circulatory half-life, and biodistribution, while reducing their side effects and immunogenicity. Furthermore, ligands such as peptides, antibodies, and nucleic acids on the surface of NPs precisely target BC cells. Studies on the synthesis of engineered NPs and their impact on BC were obtained from PubMed, Science Direct, and Google Scholar. This review provides insights on the importance of engineered NPs and their methodology for validation as a next-generation platform with preventive and therapeutic effects against BC.

## 1. Introduction

Breast cancer (BC) is the outcome of aberrant and uncontrolled cell proliferation of cancerous cells in the breast tissue. BC is the second most common cancer in females and the third-leading cause of death globally [1]. BC therapy involves a multidisciplinary approach comprising surgery as well as radiotherapy and chemotherapy as adjuvant and neoadjuvant therapies [2]. Chemotherapy is a technique that kills cancer cells using chemical agents. Although it is the most effective approach for cancer therapy, the cytotoxic effects of these chemotherapy agents generate various side effects [3]. Radiotherapy also decreases the risk of cancer recurrence and mortality. Nevertheless, it typically involves radiation exposure to adjacent organs, increasing the risk of cardiac and lung diseases. Such therapies may increase the risk of leukemia, especially in association with certain classes of adjuvant chemotherapy [4]. Conversely, these therapeutic methods are often unsuccessful in treating BC because of their adverse effects on healthy tissues and organs [5,6].

The main reason for these adverse effects and the mortality rate is the failure of therapeutic agents, which act not only on the tumor sites but also induce severe adverse effects on healthy tissues and organs, causing toxicity to the individual. BC is a highly multifaceted and heterogeneous disease and is categorized based on histopathological types. The most predominant BC cases are those of invasive ductal carcinoma, although other less-common subtypes are noteworthy due to their ferociousness and clinical manifestations [7]. The next major concern is the stage of the tumor. During cancer development, the primary tumor occurs within the breast tissue (stage 1), and then rapidly spreads to the adjacent tissues and lymph nodes (stage 2–3) or distant organs such as the lung, bone, liver, or brain (metastasis, i.e., stage 4) [7,8]. Staging of the disease is clinically important. The death rate increases as the tumor metastasizes. Moreover, BC is also categorized based on the grade and molecular subtype, viz., luminal A and B, human epidermal growth factor receptor 2 (HER2), and triple-negative BC (TNBC) [8]. Once the cancer metastasizes, the effectiveness of most standard drugs is significantly low. Finding novel, effective, and safe forms of therapy for this fatal malicious disease is thus critical. It is necessary to discover highly efficient therapeutics (the so-called “magic bullets”) which can pass through natural barriers and differentiate between benign and malignant cells in order to target malignant tissues. These agents “wisely” react to the complex tumor microenvironment for an on-demand discharge of an optimum dose range [9,10].

Tumor nanotechnology has the potential to modernize cancer diagnosis and treatment. Developments in protein engineering and material science have contributed to the development of innovative nanoscale targeting strategies, providing new optimism for BC patients. Nanoparticles (NPs), identified as pharmaceutical carriers, provide a new juncture for drug delivery to cancer cells by infiltrating tumors deeply, resulting in a high level of specificity to the targeted cancer cells [11,12,13,14,15]. Furthermore, NP treatment minimizes destructive effects on healthy tissues and organs [16,17]. Nanotechnology has been approved by the National Cancer Institute, which recognizes this technology as an outstanding paradigm-shifting approach for improving the diagnosis and treatment of BC [16].

Several therapeutic NPs, viz., Doxil^®^, Lipoplatin^®^, Onivyde^®^, Genexol-PM, and Abraxane^®^, have already been approved and are extensively employed for BC adjuvant therapy, with promising clinical outcomes [18,19,20,21]. NP-based drug delivery systems (DDSs) include several valid designs with regard to the size, shape, and nature of the biomaterials loaded with drugs, enhancing the solubility, drug stability, circulatory half-life, biodistribution, and drug release rate and reducing side effects, toxicity, and immunogenicity [22]. In this review, we provide insights into the novel design and development of engineered NPs and their payloads, which represent a tailored and promising tool for the treatment of BC. Furthermore, targeting ligands can be included on the surface of NPs, precisely targeting BC cells by attaching to the receptors on the cell surface. The tailoring of engineered NPs may have a vital role in cancer specificity, anti-drug resistance, and anti-cancerous and anti-metastasis effects.

## 2. Properties of BC Drugs

Commercial therapeutic drugs of BC can be categorized into two broad classes based on water solubility: hydrophilic (polar) and hydrophobic (non-polar). They can also be classified as highly charged or neutral drugs according to their electrostatic nature (Table 1). When engineering NPs to be employed as the carrier for a specific class of drug, it is essential to identify the behaviors and properties of the drug to achieve higher encapsulation effectiveness with the desired discharge characteristics. Functionalized or engineered NPs are highly attractive and auspicious candidates for DDS owing to their distinctive sizes, tunable surface functionalities, and well-regulated drug discharge (Figure 1).

For BC therapy, chemotherapeutic drugs are generally used either alone or in combination with other drugs. Numerous investigations have been piloted to determine the side effects of chemotherapeutic drugs both in animals and clinically [23,24,25]. For instance, the use of doxorubicin for BC results in a high possibility of complications with regard to hematopoiesis and gastrointestinal or cardiac toxicity [26,27,28,29]. Another commonly used chemotherapeutic drug is paclitaxel, which also causes several side effects, including neutropenia and peripheral neuropathy [19,30]. Similarly, other commonly used drugs for chemotherapy, such as docetaxel, cisplatin, tamoxifen, and trastuzumab, have been reported to have many side effects, including fatigue, weight loss, peripheral neuropathy, and nausea [24]. Thus, targeted delivery can be extremely essential in the treatment of BC, especially during chemotherapeutic drug usage [21,31,32].

**Table 1 pharmaceutics-13-01829-t001:** Clinical uses of anti-BC drugs.

Trade Name	Therapeutic BC Drugs	Chemical Structure	References
Nonpolar/Hydrophobic drugs
Platinol^®^	Cisplatin	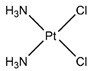	[22,33]
Taxotere	Docetaxel	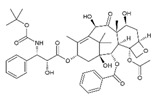	[21,31,32]
Adriamycin^®^	Doxorubicin	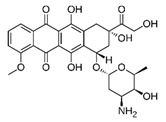	[26,27,28,29]
VP–16	Etoposide	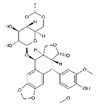	[34,35]
Otrexup™, Rasuvo^®^, Rheumatrex^®^ andTrexall™	Methotrexate	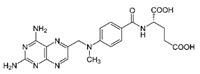	[36,37]
Taxol	Paclitaxel	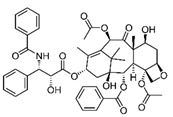	[38,39,40,41]
Polar/Hydrophilic drugs
Avastin	Bevacizumab	Monoclonal antibody	[24,42]
Erbitux^®^	Cetuximab	Monoclonal antibody	[43,44]
Cytoxan or Neosar	Cyclophosphamide	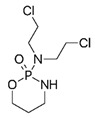	[23,24,25]
Gemzar	Gemcitabine	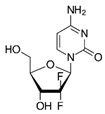	[38,40]
Adrucil^®^	5-Fluorouracil	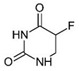	[33,45]
Zevalin	Ibritumomab	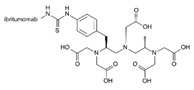	[46]
Elspar	L-asparaginase	Monoclonal antibody	[47,48]
Vectibix	Panitumumab	Monoclonal antibody	[49,50]
Rituxan	Rituximab	Monoclonal antibody	[51,52]
Bexxar	Tositumomab	Monoclonal antibody	[46,53]
Herceptin	Trastuzumab	Monoclonal antibody	[25,41]
Zolinza	Vorinostat	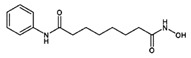	[54]
Highly charged/Neutral drugs
-	siRNA/miRNA	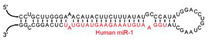	[55,56]

### 2.1. Polar/Hydrophilic Drugs

Polar/hydrophilic drugs play a vital role in treating BC subtypes, and include macromolecules (peptides, nucleic acids, polysaccharides, and proteins) and several small molecular compounds [57,58]. For instance, hydrophilic drugs such as trastuzumab (a monoclonal antibody) and gemcitabine (a nucleoside analogue) inhibit HER2 and TNBC, and thus these drugs have been used to treat early-stage and metastatic BC [25,38,40,41]. However, the effective use of hydrophilic drugs has been delayed by many setbacks, including an impaired uptake of these active drugs by cells due to their failure to cross the hydrophobic lipid-rich plasma membranes, poor bioavailability due to their low stability under enzyme digestion, and their minimal half-life in the blood circulation [57,59].

To bypass these drawbacks, NPs have been actively equipped as carriers to encapsulate and deliver potential hydrophilic drugs. Loading efficiency is the main concern to resolve because the full dosage needs to be increased when NPs with low drug contents are given [60]. During the assembly of carriers, the most commonly used materials are hydrophobic, presenting a technical hitch with regard to hydrophilic drugs owing to the deprived miscibility between these two phases. For this reason, numerous strategies have been established to improve the loading efficacy of hydrophilic drugs. For instance, drug-loading efficiency increases from 3.7%  to 47.3%  when 5-fluorouracil (5-FU) is replaced with 1-alkylcarbonyloxymethyl (an amphiphilic prodrug of 5-FU) [61]. Similarly, another study showed that adjusting the isoelectric point of a protein drug can elevate the drug-loading efficiency [62]. Electrostatic and hydrophobic interactions between lipids and a protein drug can also be increased to ease the assembly of protein–lipid complexes and hence provide greater intracellular delivery [63]. Based on these alterations, the deprived dispersion of a hydrophilic drug in NPs can result in rapid drug release. In addition, active targeting of NPs consists of the targeting moiety (e.g., antibodies, receptor ligands, or nucleic acids) trimming the surface of the nanocarrier to target receptors which are greatly upregulated in BC cells as compared to the surrounding healthy cells. For example, recent nanoformulations decorated on the surface with the anti-HER2 antibody trastuzumab and loading doses of doxorubicin or paclitaxel present elevated cellular binding as well as elevated uptake and intracellular distribution in HER2 + BC cells as compared to non-decorated NPs [64,65]. Furthermore, presently novel antibody-conjugated NPs can be active therapeutics when compared to small-molecule drug therapy. The schematic representation in Figure 1 indicates the advantages of antibody-conjugated NPs in comparison to antibody–drug conjugates. The antibodies on the NP surface can fix precisely to an overexpressed receptor on target cells, overcoming many of the limitations of nude NPs, including ineffective drug diffusion into the BC cells and the stimulation of multidrug resistance (MDR) mechanisms [65]. Antibody-conjugated NPs are the ultimate system for BC therapy and can effectively control drug loading and delivery in comparison to small molecule drug treatment.

### 2.2. Non-Polar/Hydrophobic Drugs

The most common therapeutic BC drugs presently employed in the clinic are hydrophobic, presenting continuous challenges with regard to delivery to their target. Because hydrophobic drugs are water-insoluble they are incapable of crossing the water phase (body and tissue fluids) and are unable to enter the cell membrane and intracellular targets. Moreover, intravenous treatment can also lead to clinical side effects, including embolisms and tissue toxicity [66]. Hence, an effective method to resolve the poor water solubility of hydrophobic drugs is required for encapsulation using NP-based carriers. Numerous carriers have been developed for hydrophobic drug delivery in BC therapy, including polymer micelles and polymer NPs [16]. For instance, Manatunga and co-workers found that hydroxyapatite (the mineral form of calcium apatite) in an aqueous medium can offer high drug payloads, pH sensitivity, and controlled release of combined active ingredients when encapsulated in polymer micelles [26]. NP–micelle copolymers are a carrier for DDSs which are poorly dissolved in water (e.g., paclitaxel) and considerably increase the concentration of drugs in the hydrophilic medium with encapsulation [38,39,40,41].

### 2.3. Neutral/Charged Drugs

Therapeutic BC drugs prepared based on DNA, miRNA, and siRNA are distinct types with the greatest densities of charges. They serve as an authoritative molecular therapy for BC [67]. These drugs are hydrophilic, show poor uptake by the cell, and break down faster in the physical environment. Furthermore, fast clearance occurs during systemic treatment with active drugs due to renal infiltration and the mononuclear phagocyte system. Effective delivery of these highly charged drugs via carriers is still the main hindrance to attaining therapeutic benefits. Henceforth, this approach represents the least important clinical achievement in gene therapy today [55,56].

MicroRNAs play a vital role in BC, and earlier results have also indicated that miR-21 is a promising biomarker in the diagnosis and prediction of BC [68]. Nowadays, DNA, miRNA, and siRNA drugs involve the electrostatic attraction between NPs carriers and greatly charged nucleic acids. They are negatively charged molecules under normal conditions and thus attract positively charged carriers from polymeric NPs and liposomes (which are cationic building blocks), resulting in greater loading efficiency [69,70]. Altınoglu and his colleagues established a positively charged micelle system comprising amphiphilic biopolymers, and successfully immobilized siRNA for delivery to the cell [71]. Regrettably, NPs can generate various glitches in connection with toxicity and inflammatory immune reactions [72]. The direct binding of miRNA and DNA drugs to the surface of NPs carriers is an effective strategy for BC therapy that is well established [73]. By immobilizing nucleic acids on the surface, the complications of the loading process can be reduced relative to the method of encapsulation. For instance, Chan and colleagues firstly coated Au-NPs with a layer of DNA comprising constructed sequences. Using the carrier complex of DNA-coated Au-NPs, they generated immobilizing target DNA or siRNA of interest, providing effective DDSs for BC treatment [74].

## 3. NPs for DDS

NPs generally range in size from 1 to 100 nm and have either active or passive targeting capability, with a surrounding layer of several organic or inorganic coatings that determine the properties of NPs. These properties can increase the drug concentration inside the tumor and reduce systemic toxicity in healthy tissues. Several investigations have been performed to establish the benefits of NPs in DDSs for BC therapy with regard to water dispersion, biocompatibility, biodegradability, stability, half-life in the portal circulation, renal clearance, accumulation, and uptake [11,12]. Hence, DDSs are of key importance for understanding the responses to NPs by living systems at the level of cells and tissues. For instance, liposomes are bi-layered phospholipids that can encapsulate both hydrophilic and hydrophobic drugs. Indeed, engineered liposomes can preserve the drugs until they are disturbed, showing that liposomes can promote the sustained delivery of drug formulation. Moreover, they accumulate in cancer cells and increase the selectivity of the drug function, leading to diminished toxicity [75] (Figure 2).

The enhanced retention and permeability (EPR) effect plays a key function in passively moving the NPs into BC tissues. Angiogenesis provides leaky and faulty blood vessels near the tumor spot, resulting in the EPR effect [76]. NPs, along with conventional medication, provide greater benefits due to their passive targeting. Furthermore, the treatment of NPs results in less complications and is auspicious in overwhelming MDR in tumor cells, which is the main issue in BC therapy nowadays [77]. The FDA has approved many liposomal anticancer drugs, including doxorubicin, which is a long-lasting form of encapsulated doxorubicin with liposomes that treats BC [75]. Considering this, the engineered liposome formulation contains polyethylene glycol (PEG) coated-liposomal doxorubicin, which facilitates the transport doxorubicin into tumor sites. Presently, engineered liposomal doxorubicin is employed to treat various diseases, including metastatic BC, AIDS-related Kaposi’s sarcoma, ovarian cancer, multiple myelomas, and other cancers [78].

### 3.1. In Vitro DDS

Although screening, diagnosis, and treatment methods have improved in the last decade, MDR remains a great challenge. Studies have indicated that BC resistance is generally related to several signaling pathways involving hormones, receptors, survival, apoptosis, and the stimulation of efflux pumps. The primary cause of chemotherapy failure in BC is chemoresistance. Tumor cells can trigger numerous mechanisms to escape the cytotoxic effects of drugs. Various mechanisms of MDR have been explicated, viz., changes in cell-cycle checkpoints, the loss of apoptotic processes, the restoration of injured cellular targets, and decreased accumulation of the drug. Decreased drug accumulation is due to the overexpression of one or more ATP-dependent efflux pumps, such as P-glycoprotein or mutated drug transporters [79].

The in vitro delivery of NPs is a key step in their effective function in the cell. NPs act as a drug carrier which is initially appraised at the cellular level prior to their investigation in vivo at the level of various tissues and organs. Conversely, the collection of adequate information on NP–cell interactions can allow us to tailor the properties of NPs, resulting in greater delivery in vivo and effective BC treatment (Table 2).

**Table 2 pharmaceutics-13-01829-t002:** Chemoresistance mechanisms of BC cells and their treatment by NP regimen.

Drug	Drug Uptake Pathway	Chemoresistance Mechanisms	Treatment with Nps	References
Anthracyclines	Passive diffusion	Doxorubicin-resistant MCF7 cells are more condensed, with low permeability on the plasma membrane. The overexpression of fatty acid synthase limits doxorubicin uptake through the high amount of palmitic acid in MCF7 cells. Statins reduce the lipid content and membrane rigidity	Photosensitizer nanoparticles, polyhydroxybutyrate-coated magnetic nanoparticles, and 3-aminopropoxy-linked quercetin loaded with doxorubicin have synergistic effects on a doxorubicin-resistant MCF-7 cell line	[80,81,82]
	Transporters	Overexpression of organic cation transporter 6 leads to greater resistance to doxorubicin	The loading of colchicine and coumarin-6 in oil-core carriers protects doxorubicin-resistant BC cells	[83,84]
	Endocytosis	Non-specific, adsorptive pinocytosis is increased in BC cell lines which are resistant to doxorubicin	Encapsulation of polymeric prodrug containing hyaluronic acid reduces the resistance to doxorubicin	[85]
Taxanes	Passive diffusion	Extracellular pH triggers a high migratory capacity and chemoresistance to paclitaxel and doxorubicin in MCF7 cells. The addition of cholesterol to a plasma membrane reduces paclitaxel entry into BC cells	Polymer NPs containing poly(γ-glutamic acid)-g-poly(lactic-*co*-glycolic acid) (γ-PGA-g-PLGA) loaded with doxorubicin and cholesterol-PEG form a type of combination therapy against MDR BC cells	[86]
	Endocytosis	Down-regulation of Plastin 3 increases the sensitivity of MDA-MB-231 cells to paclitaxel by an endocytosis mechanism	Surfactin loaded with doxorubicin reverses MDR BC cells	[87]
Platinum-based drugs	Passive diffusion	Levels of lipid bilayer constituents such as cholesterol, sphingomyelin, phosphatidylglycerol, and phosphatidylserine are elevated and those of phosphatidylcholine and phosphatidylethanolamines are decreased in cisplatin-resistant BC cells. Based on the membrane molecular dynamics, lipid content and cholesterol levels reduce diffusion and permeability.	Fucoidan and mesoporous platinum NPs and photothermal nanocarriers can be promising drugs for treating MDR BC cells	[88,89]

NPs normally encounter a cell and are rapidly adopted through endocytosis. Later, they transport to other cell organelles such as the nucleus, mitochondria, endosomes, lysosomes, Golgi apparatus, and endoplasmic reticulum. This transport usually facilitates the movement of vesicles along the network of the lysosome. During intracellular transport, nanocarriers undergo rapid degradation and release their payload into the organelles. Remarkably, nanocarrier surfaces can be activated with a ligand to target an exact cell organelle. Nowadays, endocytic-mediated pathways are very common, as these approaches overcome chemoresistance to drug uptake in BC cells.

For the effective delivery of BC drugs, the nanocarriers require a design enabling to them circumvent the network of the lysosome and enter the cytoplasm, which is the distinctive working site for most BC drugs. Hence, most nanocarrier drugs are designed for drug-specific targeting, cell interaction, and direct uptake into BC cells [90]. Liposomes symbolize a class of DDS vehicles that are frequently employed to disable MDR in BC treatment. Recently, epirubicin encapsulated by propylene glycol liposomes (EPI-PG-liposomes) was established as being effective in overcoming MDR in BC [91]. Another recent study has also demonstrated that engineered liposomes using arginine_8_-glycine-aspartic acid (R_8_GD) encapsulated with daunorubicin and emodin selectively deposit at the tumor site, hence demonstrating a distinct anti-BC effect [92]. Hence, the grouping of targeted engineered liposomes with chemotherapeutic drugs can lead to potential treatments for BC (Figure 3).

### 3.2. In Vivo DDS

In vitro studies on DDSs are largely focused NP–cell interactions. However, in vivo studies mainly focus on how to transfer the engineered NPs from the infusion site to the target lesion. After administration into the body, the engineered NPs reach the target cells and accrue there. The transported NPs are now connected to the immune response, with subsequent biodistribution, biodegradation, and clearance at the tissue, organ, and system levels [93]. Practically, NPs that act as the carrier of a DDS should have the following features: (a) biodegradable constituents; (b) targeting efficacy to confirm the selective accumulation at the target lesion with only minimal dosages in adjacent normal tissues/organs; (c) detoxification from the body within a short time span; and (d) features that minimize the development of resistance and immune-related noxiousness [94,95].

The most commonly employed route for therapeutic NP administration is by intravenous injection, which circumvents the obstacles in the epithelial absorption by directly entering into the blood circulation [96]. After administration, the NPs directly undergo clearance via various body systems, viz., the mononuclear phagocyte system, immune systems, kidneys, liver, spleen, and lungs [97]. Physico-chemical properties and behaviors of the NPs can potentially be affected during circulation, targeting, and clearance [98]. For instance, the smaller-sized NPs (<8 nm) can be directly filtered by the kidneys, and larger-sized NPs (8 nm and above) can either deposit in a lesion or be cleared by the mononuclear phagocyte system as they cannot undergo glomerular filtration [99,100,101].

Hydrogels are 3D, cross-linked webs of hydrophilic polymers capable of retaining water or physiological fluids in large quantities [102]. Naturally present polymers (alginate, chitosan, collagen, dextran, gelatin, and hyaluronic acid) and synthetic polymers (poly (2-hydroxyethyl methacrylate), poly (2-hydroxypropyl methacrylate), poly (ethylene oxide), poly (*N*-isopropyl acrylamide, and poly (vinyl alcohol)) are employed for the preparation of hydrogels [103]. Nowadays, tissue-engineered 3D cancer models using biomimetic hydrogels as cellular scaffolds provide an appropriate in vitro summary of the native tumor microenvironment, with huge importance for use in BC research [104]. Poly (PEG)-fibrinogen can be used as an appropriate biosynthetic hydrogel for the 3D culture of various BC cell lines, including MCF7, SK-BR-3, and MDA-MB-231. Fibrinogen-based hydrogels ease the 3D culture of BC cells and analysis of various cellular behavior in response to different characteristics of the matrix. Hence, polymeric hydrogel-based cancer models can theoretically be employed in cancer biology and anti-cancer drug-testing applications [105].

Positively charged NPs can cause systemic toxicity due to hemolysis and platelet aggregation. However, NPs are rapidly discarded by the mononuclear phagocyte system [98]. Neutral and negatively charged NPs have the longest half-life in circulation. Transforming the surface chemistry of NPs may change hydrodynamic properties, including size, surface charge, and binding capacity. Although full clearance is expected when the action is accomplished, the NPs must circumvent rapid clearance to attain the ideal targeting efficacy. Hence, the half-life of NPs in circulation should be extended to permit them to flow near the lesion many times, resulting in a higher chance of NPs being deposited at the lesion [96].

Therapeutic NPs directly pass into the circulatory system, diffusing through the vascular walls into the target lesion and discharging the payload. Owing to their huge size, NPs cannot diffuse into endothelial cells. However, intracellular vessels in the tumor site cover leaky walls, permitting NPs with the precise sizes to diffuse effectively [106]. Due to the absence or dysfunction of the lymphatic system at the tumor site, inadequate drainage generally eases the deposition of NPs in the BC tissue [107]. This phenomenon is usually achieved through EPR in passive tumor targeting. Based on this effect, both macromolecular drugs and NPs can target BC more powerfully as compared to small-molecule drugs [88,89]. The clearance of the drug with NPs normally occurs through various organs such as the liver, kidneys, lungs, spleen, as well as the complement and immune systems. This is a natural mechanism that aids in detoxifying and regulating the body. A minor quantity of intravenously administered NPs can be discarded by the blood within few minutes to hours as the force of the clearance is strong (Figure 4).

This clearance is quick with either active or passive mechanisms, resulting in a random supply between the tumor site and adjacent organs [96]. Furthermore, they can be dynamically targeted by the immune system through antibodies. Studies have shown that about one-quarter of patients with BC generate anti-PEG antibodies after treatment with PEGylated NPs [29,108,109]. The rapid clearance can be histrionically balanced with an anticipated targeting effect; however, this clearance is a noteworthy issue for the advancement of nano-drug therapy [97]. As most therapeutic BC drugs are highly noxious, they are primarily employed in the tumor region and then purged within a short time span to reduce side effects. In NP-based treatment, increased targeting efficacy and clearance of the NP carriers and an improved delivery profile of the payloads will result in in good biodistribution and pharmacokinetics. Optimal biodistribution and pharmacokinetics can be analyzed using the concentration of NPs distributed to the tumor region within a unit of time. While strategies have been achieved to enhance or reduce the deposition of NPs, there are still issues in deposition or clearance at the tumor sites [94,110,111].

Recent clinical investigations have demonstrated that age, sex, body composition, and occurrence/nonexistence of a cancer site in the breast can change the pharmacokinetics of PEGylated liposomal agents [22,55,112]. A pilot study connecting PEGylated liposomal doxorubicin and PEGylated liposomal CKD-602 (topoisomerase I inhibitor) showed that the clearance of drugs was poorer in female patients with BC aged over 60 years, possibly confirming a reduced clearance of drugs encapsulated in PEGylated liposomes [113]. Population-based pharmacokinetic studies further suggested that patients with metastatic BC have a higher clearance rate during treatment with S-CKD602 [114].

Another important subject in nanomedicine is the biodegradation of NPs in targeted tissues. NPs are normally fragmented into minor compounds and release the payload. To this end, biodegradable NPs have been established using many natural polymers such as chitosan, poly (lactic acid), poly (glycolic acid), poly (lactic-*co*-glycolic acid), gelatin, poly (alkyl cyanoacrylates), and poly (*ε*-caprolactone) [115,116]. Numerous therapeutic BC drugs, including paclitaxel, cisplatin, docetaxel, epirubicin, raloxifene, 9-nitrocamptothecin, tamoxifen, cyclophosphamide, triptorelin, and doxorubicin have been encapsulated with biodegradable NPs [20,33,117,118]. To confirm biodegradability and the therapeutic development of functional drug release, various experiments have been conducted both in vitro and in vivo [119]. These small molecules undergo catabolism in the body [99]. While they have biodegradable fragments, these NPs still exhibit certain toxic effects due to their nonspecific deposition and toxic payloads to healthy tissues [120]. Since the conception of NP therapy was conceived in 1955, various classes of nanocarrier systems have been established. Most of the NPs remain constrained to benchtop studies, and some drugs are either on the market or are presently in different phases of clinical pilot studies (Table 3).

**Table 3 pharmaceutics-13-01829-t003:** NP-based therapeutics in clinical use and under clinical investigation.

Therapeutic BC Drug	Nanocarriers	Dose and Duration	Phase of Development	BC Types	References
Paclitaxel	Albumin-bound NPs	300 mg/m^2^ for 3 weeks	Phase II	Metastatic BC	[121]
Paclitaxel	Albumin-bound NPs	100 or 125 mg/m^2^ for 1 week	Phase II	Metastatic BC	[122]
Paclitaxel	Albumin-bound NPs	260 mg/m^2^ for 3 weeks	Phase III	Metastatic BC	[117]
Paclitaxel	Albumin-bound NPs	300 mg/m^2^ for 3 weeks or 100–150 mg/m^2^ for 1 week	Phase IIb	Metastatic BC	[31]
Docetaxel	Albumin-bound NPs	100 mg/m^2^ for 1 week	Phase IIb	Metastatic BC	[31]
Paclitaxel with cyclophosphamide and trastuzumab	Albumin-bound NPs	100 mg/m^2^ for 1, 8, and 15 days	Phase II	HER2-positive BC	[23]
Paclitaxel with gemcitabine, and trastuzumab	Albumin-bound NPs	100 mg/m^2^ for 1 and 8, every 3 weeks for 6 cycles	Phase II	HER2-positive BC	[38]
Paclitaxel withpegfilgrastim	Albumin-bound NPs	260 mg/m^2^ for 3 weeks	Phase I	Metastatic BC	[39]
Paclitaxel with bevacizumab and gemcitabine	Albumin-bound NPs	150 mg/m^2^ on days 1 and 15 of a 28-day cycle	Phase II	HER2-negative metastatic BC	[40]
Paclitaxel with or without trastuzumab	Albumin-bound NPs	125 mg/m^2^ infusion weekly for 3 of 4 weeks	Phase II	HER2-positive metastatic BC	[41]
Paclitaxel with doxorubicin and atezolizumab	Albumin-bound NPs	125 mg/m^2^ for 12 weeks	Phase I	TNBC	[27]
Paclitaxel with durvalumab	Albumin-bound NPs	125 mg/m^2^ for 4 weeks	Phase II	TNBC	[123]
Paclitaxel with ipatasertib	Albumin-bound NPs	80 mg/m^2^ for 12 weeks	Phase II	TNBC	[124]
Paclitaxel with bevacizumab	Albumin-bound NPs	100 mg/m^2^ for 28 days	Phase II	TNBC	[20]
Paclitaxel with carboplatin and bevacizumab	Albumin-bound NPs	100 mg/m^2^ for 28 days	Phase III	TNBC	[20]
Paclitaxel with bevacizumab, erlotinib	Albumin-bound NPs	150 mg/m^2^ for 21 days	Phase II	TNBC	[20]
Paclitaxel with capecitabine	Albumin-bound NPs	260 mg/m^2^ for 28 days	Phase II	Locally advanced BC	[20]
Paclitaxel with grastuzumab, vinorelbine	Albumin-bound NPs	80 mg/m^2^ for 4 weeks	Phase II	Locally advanced, HER2-positive BC	[20]
Paclitaxel with carboplatin, bevacizumab, doxorubicin, cyclophosphamide	Albumin-bound NPs	150 mg/m^2^ for 4 weeks	Phase II	Locally advanced, HER2-negative BC	[20]
Paclitaxel with trastuzumab	Albumin-bound NPs	100 mg/m^2^ for 4 weeks	Phase II	Locally advanced, low HER2 BC	[20]
Paclitaxel with bevacizumab, doxorubicin, and cyclophosphamide	Albumin-bound NPs	80 mg/m^2^ for 4 weeks	Phase II	HER2-negative locally advanced BC or inflammatory BC	[24]
Doxorubicin with cyclophosphamide and mangiferin	Gold NPs	60 mg/m^2^ for 4 weeks	Phase III	Metastatic BC	[11]
Paclitaxel	Liposome	75 mg/m^2^ for 21 days	Phase III	Metastatic BC	[125]
Paclitaxel with cyclophosphamade	Liposome	60 mg/m^2^ for 21 days	Phase III	Metastatic BC	[126]
Doxorubicin with cyclophosphamide, paclitaxel, and bevacizumab	Liposome	30 mg/m^2^ for 28 days	Phase II	TNBC and ER/PR + BC	[3]
Paclitaxel	Micellar NPs	150 mg/m^2^ for 21 days	Phase II	Metastatic BC	[127]
Paclitaxel	Micellar NPs or albumin-bound NPs	260 mg/m^2^ for 3 weeks	Phase II	Metastatic BC	[128]
Doxorubicin with carboplatin	Non-PEGylated liposome	20 mg/mg/m^2^ infusion twice weekly for 3 weeks	Phase III	TNBC, HER2-positive, luminal B subtypes	[129]
Doxorubicin with cisplatin, 5-fluorouracil and trastuzumab	Non-PEGylated liposome	60 mg/m^2^ for 21 days	Phase II	ER-positive and HER2-positive BC	[33]
Doxorubicin with cyclophosphamide, docetaxel, and trastuzumab	Non-PEGylated liposome	60 mg/m^2^ for or 28 days	Phase II	ER-positive and HER2-positive BC	[25]
Cytocidal cyclin G1 construct	Pathotropic NPs	80 mg/m^2^ for 4 weeks	Phase I/II	Metastatic BC	[130]
Doxorubicin	PEGylated liposome	50 mg/m^2^ for 4 weeks	Approved	Metastatic BC	[18]
Doxorubicin	PEGylated liposome	25 mg/m^2^ for 28 days	Phase II	Metastatic BC	[131]
Doxorubicin with vinorelbine	PEGylated liposome	40 mg/m^2^ for 28 days	Phase II	Metastatic BC	[29]
Doxorubicin with gemcitabine	PEGylated liposome	25 mg/m^2^ for 3 weeks	Phase III	Metastatic BC	[132]
Doxorubicin with capecitabine	PEGylated liposome	45 mg/m^2^ for 4 weeks	Phase II	Metastatic BC	[133]
Doxorubicin with bevacizumab	PEGylated liposome	50 mg/m^2^ for 3 weeks	Phase I	Metastatic TNBC	[42]
Doxorubicin	PEGylated liposome	50 mg/m^2^ for 4 weeks	Phase II	Metastatic TNBC	[112]
Doxorubicin	PEGylated liposome	25 mg/m^2^ for 21 days	Phase I-III	Metastatic TNBC	[134]
Doxorubicin	PEGylated liposome	25 mg/m^2^ for 28 days	Phase I-III	HER2-positive BC	[135]
Paclitaxel with doxorubicin	PEGylated liposome	30 mg/m^2^ for 21 days	Phase III	Metastatic BC	[109]
Doxorubicin with trastuzumab	PEGylated liposome	40 mg/m^2^ for 28 days	Phase II	metastatic BC patients with HER2/neu over-expressing BC	[118]
Paclitaxel	Polymeric micellar NPs	300 mg/m^2^ for 4 weeks	Phase II	Metastatic BC	[136]
Paclitaxel	Polymeric micellar NPs	135–390 mg/m^2^ for 3 weeks	Phase I	Metastatic BC	[137]
Docetaxel	Polymeric NPs	20–75 mg/m^2^ for 21 days	Phase I	Metastatic BC	[32]

Antibody–drug conjugates are an evolving class of therapeutic agents that are changing the setting of targeted chemotherapy in BC. These conjugates combine the target specificity of monoclonal antibodies with the anti-cancer activity of small-molecule treatment. Numerous antibody–drug conjugates have been recently approved for the BC therapy, including brentuximab vedotin (Adcetris^®^), gemtuzumab ozogamicin (Mylotarg^®^), inotuzumab ozogamicin (Besponsa^®^), and trastuzumab emtansine (Kadcyla^®^) [138,139]. Shah and his team developed and validated a pharmacokinetic or pharmacodynamic model of antibody–drug conjugates using brentuximab vedotin, which is a model for understanding and envisaging the pre-clinical to clinical translation of antibody–drug conjugate effectiveness [140,141]. Several researchers have validated the pharmacokinetic model using in vitro and in vivo studies for antibody–drug conjugates and unconjugated drugs based on drug concentrations, preclinical tumor growth inhibition data, drug pharmacokinetics in patients, and the prediction of clinical responses [142,143,144]. Similarly, Li and his colleagues gathered the outcomes of eight clinical investigations to evaluate the ethnic sensitivity of trastuzumab emtansine and the clinically suggested dose of 3.6 mg/kg [145]. They employed different strategies to investigate the data based on comparative pharmacokinetics, non-compartmental analysis, and population-pharmacokinetic analysis of trastuzumab emtansine in Japanese patients as compared to the global population [145].

Furthermore, a retrospective analysis of inotuzumab ozogamicin was also developed and correlated preclinical and clinical pharmacokinetic or pharmacodynamic data [146]. Based on the findings, outcomes suggested that improved pharmacokinetic or pharmacodynamic models connected with therapeutic can predict drug release, exposure, and efficacy, and avert toxic or unsuccessful antibody–drug conjugates from entering or remaining on the market.

## 4. Designing of Engineered NP Carriers

Nanomedicine has the potential to evade various issues in the treatment of traditional formulations. Noteworthy strides have been made towards the application of engineered NPs for BC therapy with high sensitivity, specificity, and efficiency. The engineering of NPs primarily requires various classes of chemicals with extensive structures, sizes, and compositions [147,148]. In recent years, many techniques in nanotechnology have been developed using novel biomaterials and ligands to achieve treatments with little or no toxicity. For instance, physicochemical properties of NPs such as size, geometry or shape, composition, physical and chemical structure, charges on the surface, ligand binding, and mechanical effects can be engineered to advance their performance in vivo [101]. One fascinating example was established through the use of PEGylation, conjugation, and NP-loaded liposomes for BC diagnostics and therapeutics [149]. These techniques are feasible for reducing their deposition and toxicity in major organs to a satisfactory level by elevating their half-life in the circulation [108].

### 4.1. Organic/Inorganic Nanocarriers

Based on chemical requirements, NPs are classified as organic or inorganic. Organic NPs (macromolecular and lipid-based nanocarriers) are characterized by higher biocompatibility and biodegradability, with manifold activation and function of the drug on their surface. Inorganic nanoparticles (carbon, silica, quantum dots, and metallic NPs) exhibit high stability, with intrinsic and visual properties appropriate for theranostics. The most investigated types are metallic NPs (gold, silver, and iron oxide) that show distinctive properties (optical and electronic), and aid in cancer imaging [150]. Based on the stability pattern, therapeutic BC drugs are mostly conjugated on their surface. They can be degraded and exchange dynamics rapidly in vivo. Hybrid NPs occupy both organic and inorganic classes, improving the biocompatibility, biodegradability, and stability of the NPs (Figure 5). The application of inorganic NPs in therapy is inadequate due to their low biodegradability. Mesoporous inorganic NPs are typically biodegradable, and silica-based NPs enable us to preserve drugs within a porous morphology with physicochemical properties [151].

The large organic subfamily comprises macromolecular nanocarriers, both synthetic (polylactate derivatives, dendrimers, fluorescent organic NPs) and natural (protein, nucleic acid, ferritin, and polysaccharide-based NPs), with greater stability and several free functional groups, resulting in greater drug-loading capability [147]. Due to these functional characteristics, increasing attention is being paid to nanocarriers in BC therapy. Lipid-based NPs are also an important class in clinical investigation due to their noteworthy biocompatibility [152]. Lipid-based NPs comprise monolayer (micelles) or bilayer (liposomes) nanocarriers that can carry a wide range of materials with diverse physicochemical functions. The lipid bilayer of liposomes can be implanted with hydrophobic drugs, while hydrophilic drugs can be captured either in the aqueous core of liposomes or are exhibited on the surface [153]. Nevertheless, lipid-based NPs still have numerous issues, including instability and poor loading capacity, which lead to drug leakage. Novel hybrid NPs have been established to conjugate with subclasses. Examples include solid–lipid, hybrid polymer–lipid, and hybrid organic–inorganic NPs [154,155].

### 4.2. Natural/Synthetic Nanocarriers

Natural products are often attractive due to their abundance and higher biocompatibility, as well as the capacity adapt through biochemical mechanisms [156]. Naturally occurring substances offer many benefits over their synthetic counterparts. Natural bioactive compounds encapsulated in nanocarriers can result in increased in vivo stability and water solubility, a longer circulation time of the natural product in the blood, improved biodistribution and targeting of BC cells, and controlled and sustained drug release. They are considered potent antioxidants with closer proximity and positive effects on cancer-specific pathways, with reduced side effects [157]. Natural therapeutic drugs accumulate more appropriately for longer at the tumor site through active or passive targeting of the breast tumor tissue [158]. Several in vitro and in vivo BC model studies were established and validated the antitumor functions of nano-encapsulated phytochemicals (Table 4). Furthermore, experimental studies were established using liposomes, polymers, magnetic NPs, lipid-based NPs, and protein-based NPs, confirming the potential effects of plant-based natural products for BC therapy. The association of NPs with recognized plant-based antitumor compounds has been considered a promising method to reduce tumor growth and their adverse effects [159].

Natural materials can be rapidly metabolized and removed by the body system through hydrolytic or enzymatic degradation [158]. The most common issue with natural products is that of the immune response, which can readily occur upon administration into the body. This issue is due to the protein-derived materials; however, the response is often less severe with the administration of polysaccharide (chitosan)-derived NPs [160]. This immunogenic effect can be minimized by either chemical alteration or purification to eliminate the immunogenic constituents [161].

Presently, drug delivery takes place using NP-based-synthetic substances, as they allow an appropriate control over the physicochemical nature of the nanoproducts. NP-based synthetic substances are stable, safe, biologically inert, and are in circulation for a longer time, improving the distribution of therapeutic BC drugs to the tumor sites. The NPs can stimulate the generation of a corona of plasma proteins near the surface. Thus, extremely charged NPs are engulfed more rapidly by the mononuclear phagocyte system than neutrally charged NPs [162,163]. Hence, using synthetic nanocarriers, the hydrophobicity and surface charges of NPs can be suitably adjusted, improving their half-life in the circulation. Furthermore, their surface functions can be quickly engineered to improve their conjugation to the targeted receptors.

**Table 4 pharmaceutics-13-01829-t004:** Anti-BC effects of natural product-based nano-formulations.

Drug	Nanocarriers	Natural Compound	Size	The Outcome of the Study	BC Types	References
Doxorubicin	Poly-glycerol-malic acid-dodecanedioic acid	Curcumin	~110–218 nm	Significantly increased cytotoxicity, apoptotic cell death, and cellular intake compared to free drug in MCF-7 and MDA-MB-231	Luminal BC and TNBC	[164]
Doxorubicin	Silver NPs	Andrographolide	~450 nm	Significantly increased cytotoxicity, apoptotic cell death, and cellular intake compared to free drug in MDA-MB-453	TNBC	[158]
Adriamycin	Silver NPs	*Camellia sinensis*	~220 nm	Significantly increased cytotoxicity, apoptotic cell death, and cellular intake compared to free drug in MCF-7	Luminal BC	[165]
Doxorubicin	Folate and chitosan	Ursolic acid	~420 nm	Anticancer effects in an MCF-7 xenograft mouse model	Luminal BC	[160]
Doxorubicin	Lipid carriers (precirol^®^ ATO 5, vitamin E, poloxamer 188, Tween 80)	Sulforaphane/Isothiocyanate	145 nm	Anticancer effects in an MCF-7 xenograft mouse model	Luminal BC	[159]
Doxorubicin	Hydrophobically modified glycol chitosan with 5 beta-cholanic acid	Camptothecin	280–330 nm	Anticancer effects in an MDA-MB-231 xenograft mousemodel	TNBC	[166]
Doxorubicin	Phytosome	Quercetin	~85 nm	Anticancer effects in an MCF-7 xenograft mouse model	Luminal BC	[161]
Doxorubicin	PEGylated liposome	Gambogic acid	~107 nm	Anticancer effects in an MDA-MB-231 orthotopic xenograft mouse model	TNBC	[167]

### 4.3. Geometric Morphometry

Nanomedicine, as a discipline that involves the fields of chemistry, engineering, and material science, utilizes the unique features of NPs to design improved therapeutic BC interventions. Size, shape, density, and consistency are the key factors to be considered for not only DDSs but also for the engineering of NPs, as these factors regulate in vivo drug loading, stability, circulation, targeting capability, drug release, biodegradability, and toxicity [168]. For instance, particles that are smaller in size have a higher likelihood of accumulation during incubation and storage in vitro, and characteristically have a longer half-life in circulation in vivo [169]. Several investigations have confirmed that NPs display more benefits over micrometer-sized fragments in the range of 0.1–100 μm for DDSs [170]. The biodegradation of polymer NPs can be potentially affected by their size due to rapid degradation products.

The shape of NPs is also equally significant in the use of DDSs. Spherical NPs are generally a worthy candidate for DDSs. In addition, the morphology of anisotropy can also offer greater productivity due to their higher ratios between surface area and volume. They permit the nanocarrier to assume an encouraging shape for attachment to the cell through sharp ends and corners. Through these mechanisms NPs can cross cell membranes, and have been subjected to a wide range of investigations [171,172].

### 4.4. Surface Properties

The surface of the NPs represents another key factor determining the drug-loading efficacy, release profile, half-life in circulation, tumor targeting, and drug clearance. In fact, NPs generally have a hydrophilic surface to prevent protein adsorption and hence avoid uptake by the immune system [173]. This mechanism is generally achieved by coating the surface of NPs with a hydrophilic polymer (PEG), impacting toxicity, immunogenicity, and biodistribution [174].

The surface charge of NPs is often employed based on the zeta potential. It is determined according to the electrostatic potential of NPs, composition, and the medium used. Charged NPs with a zeta potential greater than 30 mV are indicated to be stable in suspensions, and these surface charges can generally avert the particles from aggregation. Furthermore, the surfaces of cells and blood vessels include several negatively charged ions, which resist negatively charged NPs. When the surface charge of NPs is higher, they can be hunted by the immune system, resulting in a higher clearance of NPs. Thus, the surface charge has a crucial role in minimizing the generic interactions between NPs and the immune system, averting NP loss in undesired settings. Surface hydrophilic PEG chains capsulated with NPs are frequently used to minimize generic interactions. PEGylation is considered to shield NPs such as liposomes, polymer NPs, and micelles from premature clearance during circulation. Various studies have suggested that PEGylated liposomal doxorubicin shows a prolonged half-life in the circulation and elevated stability, which is suggested to be linked to improved BC treatment efficacy [42,112,131,132,133].

Polysaccharides represent another natural surface polymer and are frequently used with NPs. They have been extensively employed in several DDSs and in tissue engineering due to their improved biocompatibility and accessibility, as well as their simple changeability. Dextran, chitosan, hyaluronic acid, fucoidan, and heparin have been employed as standard stealth-coating materials in NPs for BC therapy [86,175]. NPs coated with polysaccharides have more competent cellular uptake than other NPs due to their specific attachment with various receptors on the surface of the BC cells [176]. Hence, polysaccharides have received much attention in the field of nanomedicine.

### 4.5. Ligands

Several methods and tools are presently accessible to shield NPs for the active targeting of BC cells. Previously, monoclonal antibodies were employed to target epitopes on the cell surface; however, the widespread screening of peptide and aptamer archives has significantly extended the number of ligands available for targeted BC therapy [177]. Various ligands are presently employed, viz., antibodies, peptides, aptamers, oligosaccharides, and small molecules, and can precisely identify and attach to an overexpressed target on the surface of the cell [76,175,176,178]. This novel targeting mode involves a type of molecular recognition that initiates the binding of the ligand receptor. This conjugation allows the NPs to fix to the surface of the tumor cell selectively. Earlier studies have also confirmed this potential binding and validated these NPs as being effective in vitro and in vivo [80,81,82,83,84]. For instance, when attaching to a targeting ligand, the NPs generally demonstrate advanced internalization and are subjected to receptor-mediated endocytosis [16,43]. Based on strong conjugation with ligand, the binding affinity increases and thus promotes more effective receptor-mediated endocytosis.

Monoclonal antibodies are extensively employed as targeting ligands due to their high attaching affinity and specificity for targeting BC cells, as well as their easy accessibility. Numerous investigations have been performed using monoclonal antibodies that conjugate with all families of NPs, such as superparamagnetic iron oxide nanoparticles [179], quantum dots [180], liposomes [3], and silver nanocages [158], to contribute BC-specific targeting capacity. Using bioengineering, the monoclonal antibodies edit the redundant parts of the single-chain variable fragments, reducing the size with respect to the original antibody as well as the immunogenicity [181]. This chimeric antigen receptor-engineered T-cell is a promising tool and has extended to the treatment of other cancers, including B-cell leukemia and lymphoma [181].

Other noteworthy ligands are aptamers and peptides, which are characterized by feasible targeting methods with numerous advantages. The use of peptides shows numerous advantages, including lower molecular weight, tissue diffusion potential, loss of immunogenicity, ease of construction, and relative flexibility in chemical conjugation methods [182]. Similarly, aptamers are synthetic nucleic acid oligomers that can provide multifaceted three-dimensional structures that firmly bind to surface markers with high specificity [183]. Recently, a double aptamer–NP conjugate-based complex and adenosine triphosphate aptamer-conjugated CdTe quantum dots showed high potency for the efficient detection, monitoring, and treatment of BC [183].

Ligands such as folic acid, epidermal growth factor, and transferrin are presently more attractive for BC targeting due to their better attachment to their respective receptors with greater affinity and less immunogenicity [184,185,186]. Targeting ligands are nowadays receiving great attention due to their accessibility, assortment, high affinity, ease of attachment, and cost-effectiveness. Several ligands have been reported to conjugate with various receptors and NP families, as described in Table 5.

**Table 5 pharmaceutics-13-01829-t005:** Targeting ligands employed for BC therapy.

Type of Nps	Therapeutic BC Drug	Size of the Nps	Ligands Used for Engineering	The Outcome of the Study	BC Types	Reference
Albumin-bound NPs	2-methoxy-estradiol	~240 nm	Bovine serum albumin	Significantly enhanced cytotoxicity and cellular uptake when compared with the free drug examined in the SK-BR-3 and MCF-7 cell lines and tumor-bearing mice	HER2 + BC	[182]
Chitosan	Doxorubicin	~50 nm	Anti-HER2 peptide (5–10%) and O-succinyl chitosan graft Pluronic^®^ F127	Significantly enhanced cytotoxicity and cellular uptake when compared with the free drug in the MCF-7 cell line	HER2 + BC	[187]
Iron oxide	siRNA	130 nm	Caffeic acid, calcium phosphate, iron oxide, PEG-polyanion block copolymer	Significantly enhanced cytotoxicity and cellular uptake when compared with free drug on HCC1954. mRNA expression was decreased by 38% when compared with naked siRNA	HER2 + BC	[188]
Iron oxide	Baicalein	100 nm	PEG-coated iron oxide magnetic NPs	Significantly increased anti-apoptotic activity	TNBC	[189]
Liposome	Doxorubicin	~80 nm	1,2-Distearoyl-sn-glycero-3-phosphorylethanolamine, Distearoylphosphatidylcholine, HER2pep-K3-palmitic acid conjugate, mPEG2000	Significantly enhanced cytotoxicity and cellular uptake and reduced systemic toxicity when compared with the free drug in BT-474, SK-BR-3, and MCF-7 cell lines.	HER2 + BC	[178]
Liposome	Anti-IL6R antibody, doxorubicin	~100 nm	1,2-dioleoyl-sn-glycero-3-phosphocholine, 1,2-dioleoyl-sn-glycero-3-phosphoethanolamine, cholesterol	Significantly increased tumor-targeting efficacy with anti-tumor metastasis effects in BALB/c mice bearing 4T1 cells	Luminal BC	[190]
Liposome	Doxorubicin	194 nm	1,2-distearoyl-sn-glycero-3-phosphoryl ethanolamine, estrone conjugated dipalmitoyl phosphatidylcholine- PEG2000-NH_2_ liposomes	Significantly increased uptake in MCF-7 BC cell lines and decreased uptake in MDA-MB-231 BC cell lines	Luminal BC	[191]
PolymericNPs	Curcumin	~10 nm	Chitosan NPs with an apoptosis-inducing ligand (TRAIL)	Significantly reduced tumor volume when compared to control when tested in BALB/c mice	TNBC	[192]
Polymeric NPs	Trastuzumab	~125 nm	Antigen-binding fragments cut from trastuzumab)-modified NPs (Fab’-NPs) with curcumin	Significantly increased cytotoxicity and cellular uptake when compared with the free drug in the MDA-MB-453 cell lines and a xenograft mice model.	HER2 + BC	[193]
Polymeric NPs	Paclitaxel	~225 nm	Poly(lactic-*co*-glycolic acid) NP coated with hyaluronic acid	Significantly increased cytotoxicity and cellular uptake when compared with the free drug in MDA-MB-231.	TNBC	[194]
Polymeric NPs	Paclitaxel	131.7 nm	Hyaluronic acid-coated polyethylenimine-poly(*d*,*l*-lactide-*co*-glycolide) NPs with miR-542-3p	Significantly increased cytotoxicity and cellular uptake when compared with the free drug in MDA-MB-231.	TNBC	[195]
Polymeric NPs	Gambogic acid	121.5 nm	Hyaluronic acid-coated polyethylenimine-poly(d,l-lactide-co-glycolide) NPs with RAIL plasmid (pTRAIL) and gambogic acid	Significantly increased cytotoxicity, apoptotic cell death, and cellular uptake when compared with the free drug in MDA-MB-231.	TNBC	[196]
Polymeric NPs	Thymoquinone	~22 nm	Pluronic^®^ F127 NPs, hyaluronic acid-conjugated Pluronic^®^ P123.	Significantly reduced cell growth and migration of MDA-MB-231 cell lines and xenograft Balb/c mice	TNBC	[197]
Solid–lipid NPs	Di-allyl-disulfide	~116 nm	Pluronic F-68, solid–lipid NPs engineered with palmitic acid and soya lecithin and surface-modified with glycation end product antibodies	Significantly enhanced cytotoxicity and cellular uptake, augmented activity at the tumor site, and reduced systemic toxicity when compared with the free drug in MDA-MB231	TNBC	[198]

### 4.6. Polymeric Nanocarriers

Earlier, NPs carriers were developed and examined using a variety of materials, including monosaccharides, polysaccharides, proteins, synthetic polymers, metals, lipids, and organic/inorganic compounds. As the main prerequisite for designing NP carriers, the size, shape, composition, surface properties, and biodegradability are characteristics which must be accurately engineered and improved to achieve site-specific drug discharge with therapeutically dose-dependent optimum effects [96]. Engineered NPs activated with precise ligands can target BC cells using an appropriate method and can transport encapsulated payloads efficiently. Furthermore, advanced drug loading, enhanced half-life in the circulation, organized release, and selective delivery of NPs can also achieved by adapting the size, structure, composition, and surface properties. In the design of the engineered NPs, polymers (proteins, lipids, liposomes, dendrimers, hydrogels, organic/inorganic materials) and ligands (nucleic acids, peptides, oligosaccharides, small molecules, and antibody fragments) have been included on the surface of NPs to improve their targeting efficacy (Figure 6).

#### 4.6.1. Conjugation with Polymeric Protein

Successful DDSs are based on the attachment of the therapeutic BC drug to proteins for targeted drug delivery. These nanocarriers are directed to the BC through conjugation with the so-called antibody-conjugated NPs. These systems can protect the chemical structure of therapeutic drugs and deliver them to the BC site using a well-controlled method. Upon stimulation, the attachment of antibody to the drug is readily degradable, reducing toxicity [64]. Therapeutic choices may be limited for certain BC subtypes, and hence nanomedicine offers hope for patients with difficult-to-treat BC [199].

A major benefit arises from the smaller size (<10 nm) of such conjugates, which leads to a comparatively longer half-life in the circulation, and makes their extravasation into the BC region more successful when compared to NPs of greater sizes [46,53]. Studies have indicated that protease-cleavable conjugates are more stable than disulfides, although all of them can be engineered.

Standard chemotherapy shows low response rates and short progression-free survival among patients with pretreated metastatic TNBC. However, recent clinical studies showed that sacituzumab govitecan (an antibody–drug conjugate) is well engineered. This conjugate was heavily pre-administered to patients with metastatic TNBC. The outcomes showed improved primary endpoints with fewer complications. The secondary endpoints were progression-free and overall survival, which were found to be improved [200,201].

#### 4.6.2. Liposomes

Liposomes are spherical vesicles (ranging from 50 to 500 nm in size) with a lipid bilayer, and are generated while the lipid connects with the aqueous solution. The most commonly employed lipids are phosphatidylcholine-enriched phospholipids, which produce liposomes. They can be potentially stabilized by strengthening the bilayer with an amphiphilic, long-chain polymer holding PEG at one end, which can simultaneously decrease opsonization and lengthen the circulation time in the blood [191]. Polymeric compounds with appropriate end groups for attachments with antibodies or ligands can also be implanted into the liposome bilayer; therefore, construction-targeted delivery is conceivable. As liposomes bilayers likely mimic those of cells, they can rapidly merge with the plasma membrane [190]. When they are internalized by cells through endocytosis or passive diffusion, the lipid bilayer undergoes rapid degradation due to the acidic environment generated by the endosomes and lysosomes [178].

As shown in the illustration in Figure 2, several engineered polymeric liposomes loaded with therapeutic drugs have used for BC [29,112,135]. Passive tissue targeting is achieved by the extravasation of NPs due to the increased vascular permeability of the BC [109]. Active cellular targeting can also be achieved using the engineered liposomes of NPs with ligands that promote cell-specific recognition and binding. Based on cellular penetration, NPs can release their contents close to the BC cells [42]. Cyclic octapeptide LXY (Cys-Asp-Gly-Phe (3,5-DiF)-Gly-Hyp-Asn-Cys)-attached liposomes carrying the therapeutic drugs doxorubicin and rapamycin targeted over-expressing integrin-α3 in a TNBC-bearing mouse model [202]. These outcomes strongly indicate that targeted combinational therapy can provide a rational approach to improve the therapeutic outcomes of TNBC. Similarly, increased antitumor activity in a TNBC xenograft mouse model has also been revealed with doxorubicin and sorafenib-loaded liposomes [203]. Taken together, research on engineered polymeric liposomes suggests the potential efficacy of the drug-loaded polymer-link liposomes platform in BC therapy.

#### 4.6.3. Lipid–Hybrid Polymer

Polymer NPs are perhaps the most widely studied carrier systems targeting drug delivery. Various synthetic polymers have been employed and investigated based on the potential effects of their hydrophobic and biodegradable nature. Furthermore, many natural polymers (such as chitosan, poly (lactic acid), poly (glycolic acid), poly (lactic-*co*-glycolic acid), gelatin, poly (alkyl cyanoacrylates), and poly (*ε*-caprolactone)) have also been employed for drug delivery in BC treatment [115,116]. When liposomes and polymeric NPs were developed, a new class of lipid-hybrid NPs providing the characteristics of both systems was also established. These lipid-hybrid NP incorporate high drug-encapsulation materials and show precise drug release, with outstanding targeting capabilities. Non-targeted drug delivery of platinum–mitaplatin using poly-D, L-lactic-*co*-glycolic-acid–block-PEG NPs resulted a higher degree of tumor inhibition in the TNBC xenograft mouse model [204].

Recently, anastrozole-loaded PEGylated polymer–lipid hybrid NPs showed high entrapment efficacy (80%), size consistency, and relatively low zeta-potential values (−0.50 to 6.01), and were found to induce apoptosis in ER-positive BC cells [205]. Similarly, another study showed that nanocarriers of a polymer–lipid hybrid encapsulating psoralen optimized its hydrophilic nature and bioavailability, which improved systemic delivery [206]. Li et al. [207] developed salinomycin-loaded polymer–lipid hybrid anti-HER2 NPs and investigated the anti-tumor activity. The outcome showed that the polymer–lipid hybrid was a promising candidate targeting both HER2 + breast CSCs and BC cells.

#### 4.6.4. Dendrimers

Dendrimers are another important type of synthetic nanocarrier (with size ranges from 10 nm to 100 nm) generated by branched monomers of divergent or convergent synthesis. They appear as liposomes, showing a cavity-enriched spherical shape with a hydrophobic core and hydrophilic periphery, and are a distinctive carrier for the delivery of siRNA [208]. Wang et al. [209] established an antisense oligo attached to poly (amidoamine) dendrimers with links to the receptors of vascular endothelial growth factor, and showed a significant decrease in tumor vascularization in the TNBC xenograft mouse model. Another study reported the novel dendrimer G4PAMAM conjugated with GdDOTA and DL680, administered in TNBC xenograft mice as a model for tumor imaging and drug delivery. The outcome of MRI scan and infra-red fluorescence imaging showed the emission of NPs and a significant fluorescence signal in the tumor, demonstrating the selective delivery of a small-sized (GdDOTA)42-G4-DL680 dendrimeric agent to TNBC tumors that circumvented other adjacent primary organs [210]. Hence, the dendrimer is a potential nanocarrier and targeted diagnostic and therapeutic agent in the TNBC tumor mouse model.

## 5. Engineered NPs Increases the Circulation Half-Life

In principle, an NP-based delivery system should integrate high drug loading capability, a long circulation half-life, effective targeting capacity, discharge programmability, stimuli receptiveness, and diagnostic features. Negatively or neutrally charged NPs generally have a longer blood half-life than positively charged NPs. Using synthetic materials, the surface charges and hydrophobicity of NPs can be suitably tuned to elevate their blood half-life. Based on a longer circulation half-life, NPs can pass the lesion multiple times, with a greater chance of accruing on the lesion site [211,212]. NPs larger than 200 nm are favorably excreted by the spleen. Hence, by selecting a suitable size, surface modifications in the form of PEGylation or the use of rigid NPs will result in the longest blood half-life (2–100 folds), allowing accrual in the spleen with a high percentage and thus the improvement of overall pharmacokinetic parameters [59]. For instance, PEGylated liposomal doxorubicin exhibited a prolonged circulation half-life, which is alleged to be associated with enhanced therapeutic efficacy in BC. Studies revealed that about 50% of polystyrene NPs with the size of 250 nm and coated with poloxamine 908 accrued in the spleen almost 24 h after injection [213]. To avoid clearance by the spleen or MPS, the surface of NPs should to be cautiously engineered to avoid or at least alleviate opsonization [214].

## 6. Toxicity of NPs

NP-based drug delivery systems have reported to provide several benefits in BC treatment, including good pharmacokinetics, a precise targeting of tumor cells, reduced side effects, and MDR. Although an extensive range of NPs with diverse ions and surface alterations has been created and preclinically verified, only a limited number of drugs have gained authorization for clinical trials. Recurrent doses may cause systemic side effects, including nausea, argyria, irritation, stomach pain, allergic reactions, inflammation, and dyskinesia [215]. NP toxicity is mainly based on their base materials, size, shape, and the functional groups decorating their surface. Smaller NPs can easily diffuse into the healthy cell and interact with cellular components, including nucleic acid, proteins, and polysaccharides. Oxidative stress and ROS generation are common side effects of metal NPs, which attack all healthy cellular components and lead to cell death [216,217]. Moreover, Al_2_O_3_, CuO, Fe_3_O_4_, NiO, TiO_2_, and ZnO NPs can cause cell cycle arrest and induce apoptosis. Furthermore, NPs with positive charges have higher cytotoxicity than negatively charged NPs [218]. The shape of NPs is also largely involved in cytotoxicity in healthy cells. For instance, rod-shaped Fe_2_O_3_ NPs have greater cytotoxic effects than spherical-shaped ones [219]. Although numerous investigators have confirmed the toxicity of diverse NPs, the cause of the toxicity is largely unidentified. The short- and long-term toxicities of NPs as well as pharmacokinetic and pharmacodynamic results should be assessed prior to clinical approval.

## 7. Future Prospective

Although spherical NPs have conventionally been used for tumor targeting due to their relative ease of construction, many current studies report that non-spherical NPs such as rods, discs, hemispheres, and ellipsoids may target BC more effectively. To reach a definitive decision on the best NP shape, more detailed studies involving reticuloendothelial system clearance and cell attachment must also be performed. We must conduct investigations with ligands and receptors of relevance to BC in systems that provide an exact model of in vivo geometry, structure, and rheology. By considering the shape of NPs along with their size and material, they can be easily engineered in a manner that allows them to enter and treat BC more effectively.

Artificial intelligence (AI) is revolutionizing every area of science, medicine, and nanotechnology, including advanced BC diagnosis and treatment. This area mainly focuses on clinical images and therapies in relation to tumor size, shape, intensity, and texture, collectively leading to more comprehensive tumor characterization [220]. NP-modified drugs and imaging agents have generated improved treatment outcomes and dissimilarity efficiency. Hence, the Cancer Research UK Imperial center and the NHS foundation trust recently teamed up to improve BC diagnosis using AI [221]. The combination of image-specific findings regarding NP treatment and knowledge of the underlying genomic, pathologic, and clinical features are of great value in BC. In recent times, nanomedicine platforms have been engaged in the clinic, with authorization for Abraxane^®^ and other commercial products being bestowed. However, as with traditional/unchanged combination therapies, NP-based drug delivery is frequently explored using fixed doses.

A consistent approach for all forms of drug treatment is the use of drug combinations, which are dependent on time and dose and are patient-specific. To overcome this challenge, the evolution towards the NP-mediated co-delivery of multiple treatments has led to the potential for interfacing AI with nanomedicine for optimization in synergistic nanotherapy. AI-directed nano-robots can precisely identify the drug action at the target site of the breast through the tracking sensor. In the near future, nano-medicine research may be supported by AI, not only to diagnose the stage of cancer but also to determine potential cancer treatments [222]. Although great success has already been achieved with nanomedicine in oncological research, the use of AI in nanomedicine will be a promising solution in the future.

## 8. Conclusions

BC is the second most common cancer in females worldwide. The treatment regime for BC includes surgery, radiotherapy, and chemotherapy, which are often unsuccessful due to their various side effects. Nanomedicine has been revolutionized by allowing the exploration of new avenues for diagnosis, prevention, and therapy in BC. Therapeutic drugs or natural bioactive compounds engineered with NPs can provide ideal sizes, shapes, and charges to enhance solubility, circulatory half-life, biodistribution, and immunogenicity. Nanocarriers are engineered using organic, inorganic, natural, and synthetic approaches, involving geometric morphometrics, surface properties, ligands (peptides, antibodies, aptamers, and folic acid), and polymeric nanocarriers (protein, liposomes, lipid-hybrid, dendrimers, hydrogels). They are potentially active and target-specific, executing their role in abolishing tumor cells in the breast. Therapeutic BC drugs loaded with engineered nanocarriers enter chemoresistant cancer cells through different mechanisms, viz., endocytosis, passive diffusion, and plasma membrane transporters. The application of nanoformulations improves drug-specific targeting, cell interactions, and direct uptake into BC cells, increasing treatment efficacy. Nanomedicine-based drug delivery with engineered NPs can advance diagnostic and therapeutic outcomes, thereby contributing to increased overall survival and patient well-being.

## Figures and Tables

**Figure 1 pharmaceutics-13-01829-f001:**
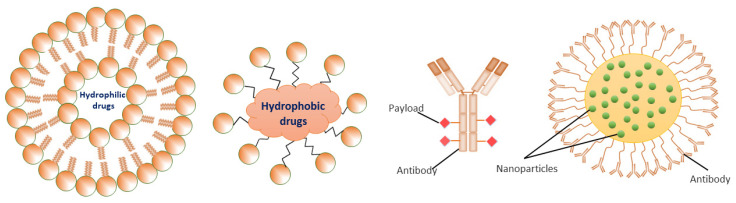
Nanocarriers used as controlled delivery vehicles for BC therapy.

**Figure 2 pharmaceutics-13-01829-f002:**
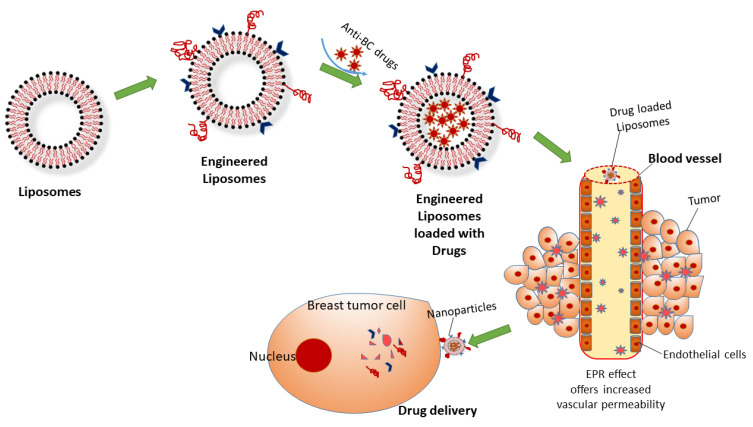
The mechanisms of engineered liposomes loaded with therapeutic BC drugs on breast tumors. Passive tissue targeting is succeeded by the extravasation of NPs through increased vascular permeability of the tumor (EPR effect). Active cellular targeting can be attained by functioned liposomes of NPs with ligands that promote cell-specific recognition and attachment. Based on the cellular penetration, NPs can release their contents close to the target cells.

**Figure 3 pharmaceutics-13-01829-f003:**
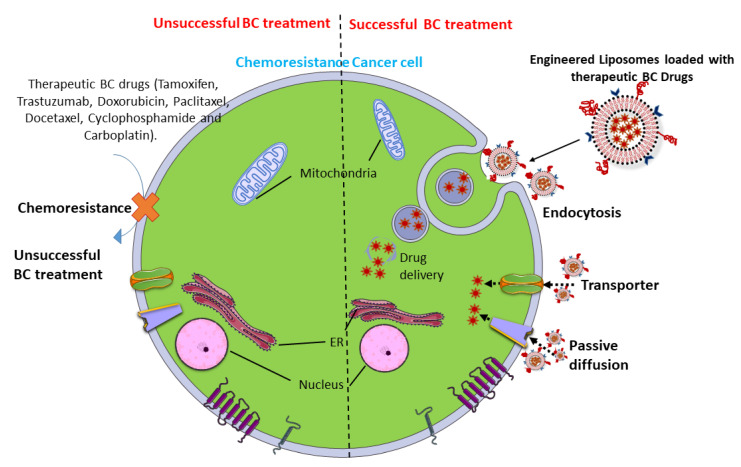
Drug uptake pathways in chemoresistant BC cells. Therapeutic BC drugs-loaded liposomes enter chemoresistant cancer cells through different mechanisms, viz., endocytosis, passive diffusion, and plasma membrane transporters. The application of nanoformulations improves drug-specific targeting, cell interaction, and direct uptake into BC cells.

**Figure 4 pharmaceutics-13-01829-f004:**
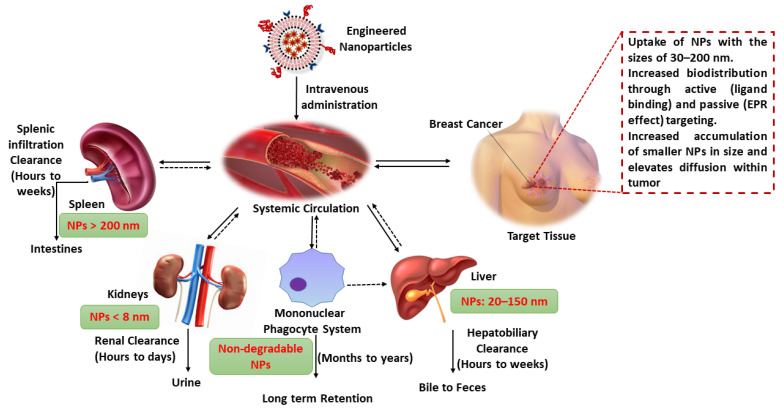
Biodistribution and clearance of NPs. NP uptake in the breast usually occurs with NP sizes of 30–200 nm; smaller-sized NPs can be diffused quicker within the targeted tumor site. The clearance of NPs is ensured through the splenic infiltration, hepatobiliary, mononuclear phagocyte, and renal systems. The solid arrows indicate direct relationships, while dashed arrows specify possible relationships. Intravenously administered NPs reach the targeted breast tumor through the systemic circulation. If the size of the NPs is smaller than 8 nm, they can be rapidly cleared within hours to days by the renal system. Large-sized non-degradable NPs are possibly taken up and recollected by the mononuclear phagocyte systems. If mononuclear phagocyte systems degrade the NPs, then they may escape sequestration and reappear in the portal circulation for renal or hepatobiliary clearance.

**Figure 5 pharmaceutics-13-01829-f005:**
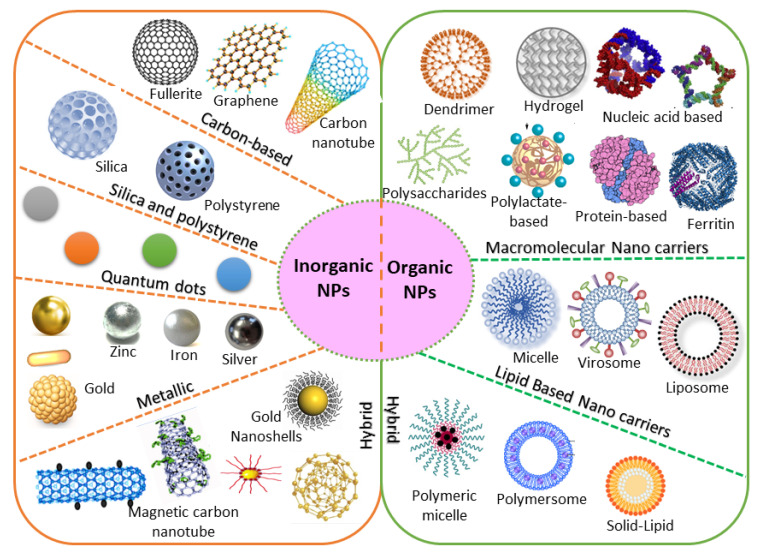
Chemical requirements for engineered NPs in BC therapy.

**Figure 6 pharmaceutics-13-01829-f006:**
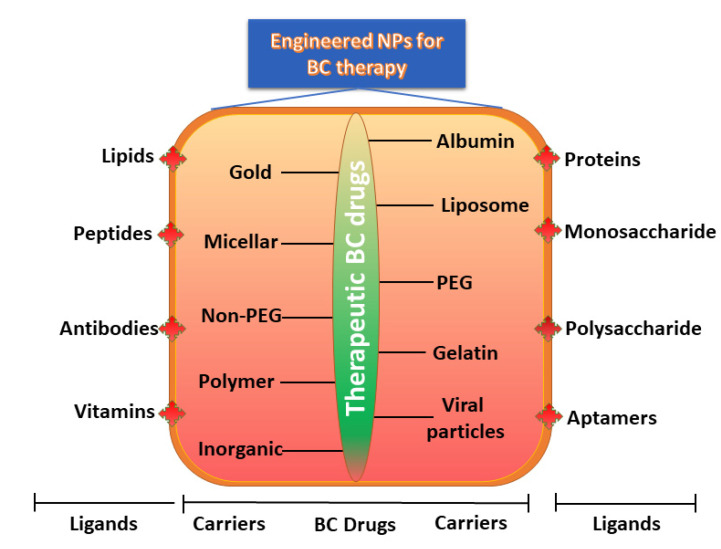
Engineered NPs for BC therapy.

## Data Availability

This study did not report any data.

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
