# Peer review of "Targeting Engineered Nanoparticles for Breast Cancer Therapy"

_pharmaceutics, 2021, doi:10.3390/pharmaceutics13111829_

Round 1

Reviewer 1 Report

This is a comprehensive review about nanoparticle-based DDS for breast cancer treatment, however there are some concerns that should be addressed by the Authors:

- Page 2 line 59-60: the sentence “Once cancer metastasizes, most of the standard drug effectiveness would be significantly low as they are not appropriate for systemic use” has to be rewritten since most of the drugs used are drugs developed for systemic administration.

- Figure 1: the figure has to be improved since the legend reports: “Nanocarriers used as controlled delivery vehicles for BC therapy” but there is no schematic representation of mAb-loaded NPs.

- Section 2.1: there is no description about the issues in the delivery of therapeutic mAb; it’s important to differentiate about the delivery of small molecule drugs and of drugs such mAb. It would be also interesting in the second part of the review (for example in section 3.2) to highlight the differences in the PK profile of small molecules or mAb delivered by NPs.

- Page 5 line 168-178: this part seems a recap of section 2 and not a part of section 2.3.

- Section 3.1: When Authors discuss about resistance, they have to highlight that the resistance that DDS can face is only that due to the overexpression of efflux pump like P-gp or mutated drug transporters.

- Table 2: the sentence “Overexpression of organic cation transporters 6 become more sensitive to doxorubicin” is correct?

- Throughout the text, some parts are redundant as for example page 9 line 288-290 is a repetition of EPR explanation; EPR was clearly explained previously.

- Section 4.6.5: This has to be modified in its position as it is not suitable for section 4.1 “Polymeric nanocarriers”, maybe it’s more appropriate in section 3.1 “In vitro DDS”.

Author Response

Comments: This is a comprehensive review about nanoparticle-based DDS for breast cancer treatment, however, there are some concerns that should be addressed by the Authors

Response: We would like to thank you for your time and effort in reviewing our manuscript, and provide your valuable suggestion to improve the quality of our manuscript. we have addressed all the suggestions given by the reviewers.

Comments: - Page 2 line 59-60: the sentence “Once cancer metastasizes, most of the standard drug effectiveness would be significantly low as they are not appropriate for systemic use” has to be rewritten since most of the drugs used are drugs developed for systemic administration.

Response: We have revised the statement according to the reviewer’s comments

Comments: - Figure 1: the figure has to be improved since the legend reports: “Nanocarriers used as controlled delivery vehicles for BC therapy” but there is no schematic representation of mAb-loaded NPs.

Response: Figure has been improved with the schematic representation of mAb-loaded NPs

Comments: - Section 2.1: there is no description about the issues in the delivery of therapeutic mAb; it’s important to differentiate about the delivery of small molecule drugs and of drugs such mAb. It would be also interesting in the second part of the review (for example in section 3.2) to highlight the differences in the PK profile of small molecules or mAb delivered by NPs.

Response: Thank you for the valuable suggestions provided by the reviewer. It has been amended according to the reviewer’s comments

Comments: - Page 5 line 168-178: this part seems a recap of section 2 and not a part of section 2.3.

Response: It has been amended according to the reviewer’s comments

Comments: - Section 3.1: When Authors discuss about resistance, they have to highlight that the resistance that DDS can face is only that due to the overexpression of efflux pump like P-gp or mutated drug transporters.

Response: It has been highlighted and included the information according to the reviewer’s comments

Comments: - Table 2: the sentence “Overexpression of organic cation transporters 6 become more sensitive to doxorubicin” is correct?

Response: it has been changed to ‘Overexpression of organic cation transporters 6 become more resistant to doxorubicin’

Comments: - Throughout the text, some parts are redundant as for example page 9 line 288-290 is a repetition of EPR explanation; EPR was clearly explained previously.

Response: It has been deleted and verified throughout the text and circumvent any repetition.

Comments: - Section 4.6.5: This has to be modified in its position as it is not suitable for section 4.1 “Polymeric nanocarriers”, maybe it’s more appropriate in section 3.1 “In vitro DDS”.

Response: As per the reviewer’s advice, section 4.6.5 has been shifted to section 3.1. in vitro DDS.

Reviewer 2 Report

The article of Kumar Ganesan et. al. entitled “Targeting Engineered Nanoparticles for Breast Cancer therapy” describes an informative study which aims to highlight the novel design and development of engineered Nanoparticles, NPs and their contribution in Breast Cancer therapy. The manuscript is highly informative as it presents the role of NPs as DDS in great detail and the way the engineered NPs are designed. Additionally, the authors describe elaborately the importance of Nanoparticles in Breast Cancer therapy. What is noteworthy to mention us that the text is well-structured, the figures and the tables are very comprehensible and the future perspectives of this research are correctly subjoined in the text. However, there are several issues, which are referred below:

  • The authors should give additional information regarding the half-life of NPs. It is highly recommended that the factors determining half-life are referred in the text.
  • In the manuscript, the authors should add a paragraph referring to the possible side effects NP-based drugs could have, if delivered to an organism.

Author Response

Comments: The article of Kumar Ganesan et. al. entitled “Targeting Engineered Nanoparticles for Breast Cancer therapy” describes an informative study which aims to highlight the novel design and development of engineered Nanoparticles, NPs and their contribution in Breast Cancer therapy. The manuscript is highly informative as it presents the role of NPs as DDS in great detail and the way the engineered NPs are designed. Additionally, the authors describe elaborately the importance of Nanoparticles in Breast Cancer therapy. What is noteworthy to mention us that the text is well-structured, the figures and the tables are very comprehensible and the future perspectives of this research are correctly subjoined in the text.

Response: Thank you for your positive comments and encouragement.

Comments: However, there are several issues, which are referred below:

Response: We thank the reviewer for their valuable time and effort in reviewing our manuscript, and for your valuable suggestion to improve the quality of our manuscript

Comments: The authors should give additional information regarding the half-life of NPs. It is highly recommended that the factors determining half-life are referred in the text.

Response: It has been highlighted in section 5 according to the reviewer’s comments

Comments: In the manuscript, the authors should add a paragraph referring to the possible side effects NP-based drugs could have, if delivered to an organism.

Response: It has been highlighted in section 6 according to the reviewer’s comments

Reviewer 3 Report

Ganesan and co-workers elaborated an extensive review with more than 200 references, most of them published within the last 5 years.  The authors are recommended to revise the manuscript prior a decision to publish.

The title does not reflect the amplitude of the work, focusing only on "Targeting Engineered Nanoparticles for Breast Cancer therapy" that is explored only in section 4. The manuscript contains many information, not always organized in a clear and simple way. For example Table 3 (recommended to organize in order of BC drug). Also, the description of NPs types is dispersed in the manuscript, it would be better to organize them more systematically.

In the resume, the literature search tools are listed, but no information regarding the number of references found is mentioned.

The figures are very well designed.  Yet, the title should reflect the content of the review or the review need to be revised to focus only in the title topic.

Author Response

Comments: Ganesan and co-workers elaborated an extensive review with more than 200 references, most of them published within the last 5 years.  The authors are recommended to revise the manuscript prior a decision to publish.

Response: Thank you for your positive comments and encouragement

Comments: The title does not reflect the amplitude of the work, focusing only on "Targeting Engineered Nanoparticles for Breast Cancer therapy" that is explored only in section 4. The manuscript contains many information, not always organized in a clear and simple way. For example Table 3 (recommended to organize in order of BC drug). Also, the description of NPs types is dispersed in the manuscript, it would be better to organize them more systematically.

Response: Based on the review comments, the work is greatly focused only on Engineered Nanoparticles for Breast Cancer therapy. We have done our best to organize the manuscript in a clear way. Table 3 is now organised NPs-based BC therapeutics in clinical use and under clinical investigation. In addition, We organized NPs types in the manuscript systematically.

Comments: In the resume, the literature search tools are listed, but no information regarding the number of references found is mentioned.

Response: As a narrative review, we don’t search systematically using inclusion and exclusion criteria based on the literature search tools. Hence, We haven’t searched whether full paper/ abstract, countrywide, or how many references are available in PubMed, Science Direct, and Google Scholar. We have done our top priority to incorporate recent  information on Engineered Nanoparticles for Breast Cancer therapy

Comments: The figures are very well designed.  Yet, the title should reflect the content of the review or the review need to be revised to focus only in the title topic.

Response: It has been amended according to the reviewer’s comments

Round 2

Reviewer 1 Report

The article is now suitable for publication in Pharmaceutics